**Data Availability Statement:** All data files are available from the VAERS database: https://vaers.hhs.gov/data.html.

# Reported rates of all-cause serious adverse events following immunization with BNT-162b in 5–17-year-old children in the United States

**Halinder S. Mangat**[1]*, **Brady Rippon**[2], **Nikita T. Reddy**[3], **Akheel A. Syed**[4], **Joel M. Maruthanal**[1], **Susanne Luedtke**[5], **Jyothy J. Puthumana**[6], **Abhinash Srivatsa**[7], **Arnold Bosman**[8], **Patty Kostkova**[9]

1 Department of Neurology, University of Kansas Medical Center, Kansas City, Kansas, United States of America, 2 Department of Population Health Sciences, Weill Cornell Medical College, New York, New York, United States of America, 3 Department of Medical Sciences, Newcastle University, Great Britain, Newcastle upon Tyne, United Kingdom, 4 Faculty of Biology Medicine and Health, The University of Manchester, Manchester, United Kingdom, 5 Gesundheitsamt Nuremberg, Nuremberg, Germany, 6 Department of Cardiology, Northwestern Medicine, Chicago, Illinois, United States of America, 7 Department of Pediatrics, Boston's Children's Hospital, Boston, Massachusetts, United States of America, 8 Transmissible BV, Public Health Learning Solutions, Utrecht, The Netherlands, 9 UCL Centre for Public Health in Emergencies (dPHE), University College London, London, United Kingdom

* hmangat@kumc.edu

## Abstract

Vaccine development against COVID-19 has mitigated severe disease. However, reports of rare but serious adverse events following immunization (sAEFI) in the young populations are fuelling parental anxiety and vaccine hesitancy. With a very early season of viral illnesses including COVID-19, respiratory syncytial virus (RSV), influenza, metapneumovirus and several others, children are facing a winter with significant respiratory illness burdens. Yet, COVID-19 vaccine and booster uptake remain sluggish due to the mistaken beliefs that children have low rates of severe COVID-19 illness as well as rare but severe complications from COVID-19 vaccine are common. In this study we examined composite sAEFI reported in association with COVID-19 vaccines in the United States (US) amongst 5-17-year-old children, to ascertain the composite reported risk associated with vaccination. Between December 13, 2020, and April 13, 2022, a total of 467,890,599 COVID-19 vaccine doses were administered to individuals aged 5–65 years in the US, of which 180 million people received at least 2 doses. In association with these, a total of 177,679 AEFI were reported to the Vaccine Adverse Event reporting System (VAERS) of which 31,797 (17.9%) were serious. The rates of ED visits per 100,000 recipients were 2.56 (95% CI: 2.70–3.47) amongst 5-11-year-olds, 18.25 (17.57–18.95) amongst 12-17-year-olds and 33.74 (33.36–34.13) amongst 18-65-year olds; hospitalizations were 1.07 (95% CI 0.87–1.32) per 100,000 in 5-11-year-olds, 6.83 (6.42–7.26) in 12-17-year olds and 8.15 (7.96–8.35) in 18–65 years; life-threatening events were 0.14 (95% CI: 0.08–0.25) per 100,000 in 5-11-year olds, 1.22 (1.05–1.41) in 12-17-year-olds and 2.96 (2.85–3.08) in 18–65 year olds; and death 0.03 (95% CI 0.01–0.10) per 100,000 in 5–11 year olds, 0.08 (0.05–0.14) amongst 12-17-year olds and 0.76 (0.71–0.82) in 18–65 years age group. The results of our study from national population surveillance data demonstrate rates of reported serious AEFIs amongst 5–17-

**Funding:** The author(s) received no specific funding for this work.

**Competing interests:** The authors have declared that no competing interests exist.

**Abbreviations:** Ad26.COV2.S, Janssen COVID-19 Vaccine; BNT-162b, Pfizer-BioNTech COVID-19 Vaccine; CI, Confidence Interval; COVID-19, coronavirus disease 2019; ED, Emergency Department; IRR, Incidence rate ratio; mRNA, messenger ribonucleic acid; mRNA-1723, Moderna COVID-19 Vaccine; sAEFI, serious adverse event following immunisation; SARS-CoV-2, Severe acute respiratory syndrome coronavirus 2; VAERS, Vaccine Adverse Events Reported Systems.

year-olds which appear to be significantly lower than in 18-65-year-olds. These low risks must be taken into account in overall recommendation of COVID-19 vaccination amongst children.

## Introduction

COVID-19 has caused twice as many hospitalizations amongst unvaccinated 5–11-year-olds and 4-times as many in 12–17-year-olds compared to age-matched vaccinated children [1]. Despite wide availability of vaccines, reports of rare but serious adverse events following immunization (sAEFI) have caused **significant vaccine hesitancy** amongst parents, with 42–66% reluctant or opposed to vaccination in children [2]. While myopericarditis following vaccination with mRNA vaccines has been reported in young male recipients, causing vaccine hesitancy, the absolute incidence of myopericarditis is 2–6 times lower than in those suffering COVID-19 [3]. This study examines the composite all-cause rates of sAEFI associated with COVID-19 vaccine amongst 5–17-year-olds recipients and **reported** to the Vaccine Adverse Events Reported System (VAERS) surveillance registry to estimate COVID-19 vaccine risks amongst children and compare these to adults.

## Materials and methods

In this observational study, we analyzed data on sAEFI viz. hospitalizations, life-threatening events, and deaths reported to the VAERS following COVID-19 vaccination amongst 5-17-year-old and compared them to adults 18-65-year-olds between December 13, 2020, and April 13, 2022. All events that occurred up to 42 days after vaccination were included given the time-lags in manifestation of cerebral venous sinus thrombosis. The study was exempt from institutional ethical review per institutional requirements, as it is secondary analysis of publicly available and anonymized data. No consent was required for the collection of data.

VAERS is a voluntary adverse event reporting system for all vaccines administered to children or adults, established by the Center for Disease Control and Prevention (CDC) [4]. Healthcare providers are required to report any listed adverse event from the VAERS 'Table of Reportable Events', such as hospitalization, life-threatening event, death, permanent disability, congenital anomaly, or birth defect that occurs following vaccination within a pre-specified time-period, or any similar adverse event listed as a contraindication to further doses of the vaccine. The VAERS registry includes data on demographics, geographical location, date(s) of vaccination, date(s) of adverse event report, symptoms, recovery, disability, and if there is a report that any healthcare was sought; all entries are anonymized, and data is publicly accessible. Unlike absolute risks, sAEFI rates in VAERS are subject to biases. As stated above, whilst reporting rates of all AEFI range widely (28–72%), sAEFI are more accurately recorded by physicians and reported in hospitals, compared to minor AEFIs seen in primary care [5]. We have previously used similar methodology to report on incidences of reported sAEFIs amongst adults [6].

### Exposure

The primary exposures of interest were SARS-CoV-2 vaccines: BNT-162b2 (Comirnaty, Pfizer-BioNTech) for children and BNT-162b2, mRNA-1273 (Spikevax, Moderna) and Ad26.COV2.S (Jcovden, Janssen) for adults. National vaccine administration demographics and vaccine manufacturer data were obtained from the CDC public access portal [7, 8].

## Outcomes

We focused on sAEFI viz. hospitalizations, life-threatening events, and deaths attributed to the SARS-CoV-2 vaccines due to population level implications. In addition, we included emergency department (ED) visits to determine whether increased visits to ED were related to sAEFI. These four healthcare outcomes are also less likely to be underreported; adverse events serious enough to warrant a hospital visit are mandated to be reported to VAERS [9].

## Statistical analysis

The VAERS dataset for all AEFI attributed to SARS-CoV-2 vaccines was downloaded, reformatted, and restricted to vaccines administered between December 13, 2020, and April 13, 2022. Duplicate entries, those with missing vaccination date or manufacturer information were excluded. Data on numbers of 1$^{st}$, 2$^{nd}$ and booster vaccine doses administered were available, but the adverse events are not reported by dose number; therefore, it was not possible to calculate reported event rates per persons or dose sequence, but rather per total doses administered.

## Determination of cumulative reporting rates

Cumulative reporting rates of each reported outcome were calculated for each vaccine and vaccine type. Rates were calculated as cumulative reported sAEFI per 100,000 administered doses for the period for each vaccine, and 95% confidence intervals (CI) were generated. Additional descriptive analyses included the generation of graphical outputs of temporal trajectories of moving 7-day averages of sAEFI for the three vaccines to visualize timelines of reporting rates.

## Comparing relative rates for sAEFI reporting between vaccines

A generalized Poisson regression model was used to calculate reporting incidence rate ratios (IRR) for each of the four outcomes (i.e., ED visits, hospitalizations, life-threatening events, and death) with 95% CIs. The model was adjusted for age [grouped as 5–11, 12–17, 18–65 (referent category)] and sex [males and females (referent category)]. All data management and formatting were carried out in Stata 17 (StataCorp, College Station, TX, USA). All statistical analyses were performed in RStudio (1.4.1717).

## Results

A total of 467,890,599 vaccine doses were administered to individuals aged 5–65 years and a total of 180,581,278 individuals were fully vaccinated (with 2-doses of mRNA and 1 dose of Janssen) with any vaccine (Table 1). BNT-162b2 was the sole vaccine approved for the 5–17-year-olds during the period of this study though very rare sAEFI were also reported to mRNA-1273 and Ad26.COV2.S possibly from off-label use.

Amongst all recipients, AEFIs attributed to BNT-162b totaled 177,679 and sAEFI were 31,797. The overall crude cumulative reported rate of sAEFI per 100,000 fully vaccinated was lowest for 5–11 years age group [3.06 (95% CI 2.70–3.47)] followed by 12–17 [18.25 (95% CI 17.57–18.95)] and 18–65 years [33.74 (95% CI 33.36–34.13)] (Table 2). The crude cumulative reported rates per 100,000 fully vaccinated 5–11-year-olds for ED visits were 2.56 (95% CI 2.23–2.94), hospitalizations 1.07 (95% CI 0.87–1.32), life-threatening events 0.14 (95% CI 0.08–0.25) and death 0.03 (0.01–0.10). The respective rates for 12–17-year-olds for ED visits were 14.39 (95% CI 13.79–15.01), hospitalization 6.83 (95% CI 6.42–7.26), life-threatening events 1.22 (95% CI 1.05–1.41) and death 0.08 (95% CI 0.05–0.14) (Fig 1).

**Table 1. Descriptive characteristics of the cohort.**

|  | All SARS-CoV2 N(%) | BNT-162b2 n (%) | mRNA-1273 n (%) | Ad26.COV2.S n (%) |
|---|---|---|---|---|
| **Fully vaccinated (%)** | 180,581,278 | 108,226,689 (59.93) | 57,629,041 (31.91) | 14,725,548 (8.15) |
| **Age category** (years) |  |  |  |  |
| 5–11 | 3,844 (1.00) | 3,791 (2.13) | 50 (0.03) | 3 (0.01) |
| 12–17 | 14,856 (3.85) | 12,455 (7.01) | 1,978 (1.13) | 423 (1.29) |
| 18–65 | 367,505 (95.16) | 161,433 (90.86) | 173,664 (98.85) | 32,408 (98.70) |
| **Sex** |  |  |  |  |
| Female | 276,780 (71.67) | 125,136 (70.43) | 131,003 (74.56) | 20,641 (62.86) |
| Male | 106,782 (27.65) | 51,218 (28.83) | 43,485 (24.75) | 12,079 (36.79) |
| Missing | 2,643 (0.68) | 1,325 (0.75) | 1,204 (0.69) | 114 (0.35) |
| **Any Adverse Events (%)** | 386,205 | 177,679 (46.01) | 175,692 (45.49) | 32,834 (8.50) |

Characteristics of patients with reported serious adverse events within 42 days of receiving SARS-CoV2 vaccination between December 13, 2020, and April 13, 2022, inclusive. (Data was obtained from the vaccine adverse events reported system (VAERS) registry in the United States). (BNT-162b2 –Pfizer-Biontech; mRNA-1273 – Moderna; Ad26.COV2.S –Janssen)

Compared to 18–65-year-olds, IRRs in 5–11 and 12–17-year groups were lower for all reported sAEFI except for ED visits in 12–17-year-olds [IRR 1.08 (95% CI 1.04–1.13)] and hospitalizations [IRR 1.60 (95% CI 1.50–1.70)] (Table 3).

Males in the whole cohort had higher reported rates for sAEFI. IRRs for age-sex interactions (reference group 18–65-year-old females) showed a higher reported rate of hospitalization for 12–17-year-old males [IRR 1.43 (95% CI 1.24–1.65)].

## Discussion

These data demonstrate that overall reported rates of sAEFI amongst children following BNT-162b vaccination in the United States are very low. However, compared to adults, 12–17-year-olds appear to have a higher rate of reported ED visits and hospitalizations, but this did not translate into increase in reported life-threatening events or deaths. The main related clinical diagnosis reported has been myopericarditis, with an estimated incidence of 17/100,000 doses. Of those hospitalized, 96% were only for observation, with 98% discharged home, and very rare deaths [10]. These rates closely mirror both the overall crude rates of reported hospitalizations from VAERS (6.8/100,000 fully vaccinated (2 doses), i.e., 14 per 100,000 doses), as well as rates of composite all-cause life-threatening events and deaths in the surveillance data. Reported sAEFI rates from these post-marketing safety surveillance data are reassuring in that the nationwide data on all-cause sAEFI are limited to hospitalizations (for observation in myopericarditis) but extremely rare life-threatening events or deaths in this age group. We have demonstrated similarly low rates amongst adult age-groups as well [6].

These data are useful to public health officials to reassure anxious parents and adolescents and help enhance vaccine trust. Of the reported 50% of parents willing to have their children receive SARS-CoV-2 vaccine, only about one-fifth reported that they will actually vaccinate their children within 3 months of their eligibility [11]. Additionally, children who test positive for SARS-CoV-2 via nucleic acid test are more likely to have post-COVID-19 conditions such as systemic symptoms, fever, lethargy, anorexia, myalgia, edema, respiratory symptoms, chest pain, gastrointestinal symptoms, neurological symptoms, or rashes up to 90–120 days after an ED visit [12]. Meanwhile vaccination was associated with preventing ED and urgent care visits among 12-17-year-olds during both Delta and Omicron waves, 89% and 73%, respectively [13] while protecting against severe illness and hospitalization [14].

**Table 2. Crude cumulative reported rates for serious adverse events associated with BNT-162b2 amongst over 5-year-old vaccinated recipients.**

| | Total reported events n (%) | Crude cumulative 42-day reported rate (95% CI) |
|---|---|---|
| **Any reported adverse event** | **177,679** | |
| 5–11yrs | 3,791 (2.13) | 47.78 (46.28–49.32) |
| 12–17yrs | 12,455 (7.01) | 84.36 (82.90–85.86) |
| 18–65yrs | 161,433 (90.86) | 188.75 (187.83–189.67) |
| **Any reported serious adverse event** | **31,797 (17.90)** | **29.38 (29.06–29.70)** |
| 5–11yrs | 243 (0.76) | 3.06 (2.70–3.47) |
| 12–17yrs | 2,694 (8.47) | 18.25 (17.57–18.95) |
| 18–65yrs | 28,860 (90.76) | 33.74 (33.36–34.13) |
| **ED visit** | **27,774 (15.63)** | **25.66 (25.36–25.97)** |
| 5–11yrs | 203 (0.73) | 2.56 (2.23–2.94) |
| 12–17yrs | 2,124 (7.65) | 14.39 (13.79–15.01) |
| 18–65yrs | 25,447 (91.62) | 29.75 (29.39–30.12) |
| **Hospitalization** | **8,067 (4.54)** | **7.45 (7.29–7.62)** |
| 5–11yrs | 85 (1.05) | 1.07 (0.87–1.32) |
| 12–17yrs | 1,008 (12.50) | 6.83 (6.42–7.26) |
| 18–65yrs | 6,974 (86.45) | 8.15 (7.96–8.35) |
| **Life-threatening event** | **2,723 (1.53)** | **2.52 (2.42–2.61)** |
| 5–11yrs | 11 (0.40) | 0.14 (0.08–0.25) |
| 12–17yrs | 180 (6.61) | 1.22 (1.05–1.41) |
| 18–65yrs | 2,532 (92.99) | 2.96 (2.85–3.08) |
| **Death** | **667 (0.38)** | **0.62 (0.57–0.66)** |
| 5–11yrs | 2 (00.30) | 0.03 (0.01–0.10) |
| 12–17yrs | 12 (1.80) | 0.08 (0.05–0.14) |
| 18–65yrs | 653 (97.90) | 0.76 (0.71–0.82) |

Crude cumulative reported rate of serious adverse events occurring within 42 days after SARS-CoV2 vaccination, amongst all recipients of BNT-162b2 recipients above the age of 5 years reported to VAERS between December 13, 2020, and April 13, 2022. Reported rate is expressed per 100,000 patients fully vaccinated with BNT-162b2. CI–confidence intervals; ED–emergency department

## Limitations

The raw data was obtained from VAERS, a national vaccine safety surveillance database, which is limited by reporting and information biases thus may not accurately assess the absolute incidence of sAEFI but may provide early data on actual reports of sAEFI [4]. However, serious AEFI are more likely to occur in hospitals given they involve hospitalization, life-threatening events, or death. And hospital-based physicians have demonstrated greater reported rates than community physicians who may primarily diagnose milder AEFIs in their outpatient practice [5]. That is why we selected serious adverse events and anticipate greater accuracy in their reporting.

## Conclusion

Rates of reported sAEFI after SARS-CoV-2 vaccination with BNT-162b2 appear to be very low amongst 5–17-year-old children, particularly when compared to adults. Vaccine-related hospitalizations were reported more often in 12–17-year-old males but did not appear to progress to

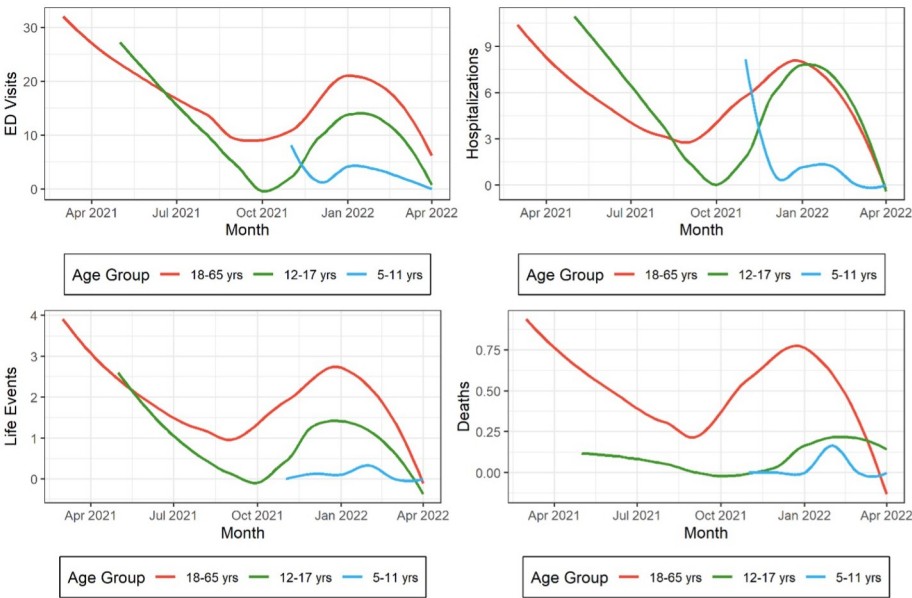

**Fig 1. Monthly crude cumulative reported rate of sAEFI types occurring within 42 days after SARS-CoV2 vaccination among recipients of BNT-162b in adults (>17 years), and 5–11 and 12–17-year-old children.** Reported rates are expressed per 100,000 patients fully vaccinated. Temporal trends have been smoothed using local polynomial regression lines. A slight increase in all rates reflects the concomitant delta wave in US during winter 2021. The scale is reduced (0–4 per 100,000 & 0–1 per 100,000 for life-threatening events and deaths respectively) in insets to demonstrate minimal risk increase of these events concurrently as well.

life-threatening events or death, but the overall reported rates were very low. The low rates of reported sAEFI suggest that BNT-162b2 vaccination is safe in children. As COVID-19 becomes endemic and the cumulative risk of long-COVID increases, safety data from surveillance databases such as VAERS are helpful, and must be utilized to assuage parental anxiety, assist primary care physicians to adequately inform their patients, and for public health officials to promote trust in vaccines. Accurate and timely articulation of these adverse events data are paramount to dispelling misinformation or confusion.

**Table 3. Incidence rate ratios for reported outcomes.**

|  | ED visit | Hospitalisation | Life-threatening events | Death |
|---|---|---|---|---|
|  | IRR (95% CI) | IRR (95% CI) | IRR (95% CI) | IRR (95% CI) |
| **Age group** (years) |  |  |  |  |
| 18–65 (referent) | 1 | 1 | 1 | 1 |
| 12–17 | 1.08 (1.04–1.13) | 1.60 (1.50–1.70) | 0.79 (0.68–0.92) | 0.17 (0.10–0.31) |
| 5–11 | 0.34 (0.30–0.39) | 0.43 (0.35–0.54) | 0.16 (0.09–0.28) | 0.09 (0.02–0.37) |
| **Sex** |  |  |  |  |
| Female (referent) | 1 | 1 | 1 | 1 |
| Male | 1.02 (0.99–1.04) | 2.10 (2.01–2.19) | 1.97 (1.82–2.12) | 4.28 (3.66–5.01) |
| Missing | 0.40 (0.32–0.49) | 0.46 (0.31–0.70) | 0.77 (0.44–1.35) | 4.57 (2.50–8.33) |

Multivariable Generalised Poisson regression model exploring association between the age and outcomes among recipients of BNT-162b2, adjusted for sex. (December 13, 2020, to April 13, 2022, inclusive). IRR–Incidence Rate Ratio; CI–confidence intervals; ED–emergency department.

## Author Contributions

**Conceptualization:** Halinder S. Mangat, Akheel A. Syed, Susanne Luedtke, Arnold Bosman.

**Data curation:** Halinder S. Mangat, Brady Rippon.

**Formal analysis:** Brady Rippon, Patty Kostkova.

**Investigation:** Halinder S. Mangat.

**Methodology:** Halinder S. Mangat, Brady Rippon, Akheel A. Syed, Patty Kostkova.

**Project administration:** Halinder S. Mangat.

**Resources:** Nikita T. Reddy.

**Supervision:** Susanne Luedtke, Patty Kostkova.

**Validation:** Brady Rippon.

**Visualization:** Brady Rippon.

**Writing – original draft:** Halinder S. Mangat, Nikita T. Reddy, Akheel A. Syed, Joel M. Maruthanal, Susanne Luedtke, Jyothy J. Puthumana, Abhinash Srivatsa, Arnold Bosman, Patty Kostkova.

**Writing – review & editing:** Halinder S. Mangat, Brady Rippon, Nikita T. Reddy, Akheel A. Syed, Joel M. Maruthanal, Susanne Luedtke, Jyothy J. Puthumana, Abhinash Srivatsa, Arnold Bosman, Patty Kostkova.

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
