## [Decision Letter · Decision Letter 0]

7 Feb 2023

Reported Rates of All-Cause Serious Adverse Events Following Immunization with BNT-162b in 5–17-Year-Old Children in the United States

PONE-D-22-34028

Dear Dr. Mangat,

We’re pleased to inform you that your manuscript has been judged scientifically suitable for publication and will be formally accepted for publication once it meets all outstanding technical requirements.

Kind regards,

Marwa Shawky Abdou, DPH

Academic Editor

PLOS ONE

1. Please note that in order to use the direct billing option the corresponding author must be affiliated with the chosen institute. Please respond by return e-mail so that we can amend your submission or remove this option. We can make any changes on your behalf.

Additional Editor Comments: Please construct abstract more specifically by background, methods, results, and conclusion

Reviewers' comments:

Reviewer's Responses to Questions

**Comments to the Author**

1. Is the manuscript technically sound, and do the data support the conclusions?

Reviewer #1: Yes

2. Has the statistical analysis been performed appropriately and rigorously? 

Reviewer #1: Yes

3. Have the authors made all data underlying the findings in their manuscript fully available?

Reviewer #1: Yes

4. Is the manuscript presented in an intelligible fashion and written in standard English?

Reviewer #1: Yes

5. Review Comments to the Author

Reviewer #1: This article describes reports of serious adverse events submitted to the US adverse event surveillance system VAERS. It is a straightforward summary of publicly available data, stratified to provide comparison across 3 age categories. Conclusions are drawn on the lower reporting of serious adverse events in children and adolescents compared to adults as a reference group. The paper is clearly written and provides appropriate analysis and conclusions.

6. PLOS authors have the option to publish the peer review history of their article (what does this mean?). If published, this will include your full peer review and any attached files.

Reviewer #1: No

---

## [Editor Report · Acceptance letter]

9 Feb 2023

PONE-D-22-34028 

Reported Rates of All-Cause Serious Adverse Events Following Immunization with BNT-162b in 5–17-Year-Old Children in the United States 

Dear Dr. Mangat:

I'm pleased to inform you that your manuscript has been deemed suitable for publication in PLOS ONE. Congratulations! Your manuscript is now with our production department. 

Kind regards, 

on behalf of

Dr. Marwa Shawky Abdou 

Academic Editor

PLOS ONE